# Toward Modeling Psychomotor Performance in Karate Combats Using Computer Vision Pose Estimation

**DOI:** 10.3390/s21248378

**Published:** 2021-12-15

**Authors:** Jon Echeverria, Olga C. Santos

**Affiliations:** 1Computer Science School, Universidad Nacional de Educación a Distancia (UNED), 28040 Madrid, Spain; 2aDeNu Research Group, Artificial Intelligence Department, Computer Science School, Universidad Nacional de Educación a Distancia (UNED), 28040 Madrid, Spain; ocsantos@dia.uned.es

**Keywords:** human activity recognition (HAR), computer vision, deep learning, human pose estimation (HPE), OpenPose, martial arts, karate

## Abstract

Technological advances enable the design of systems that interact more closely with humans in a multitude of previously unsuspected fields. Martial arts are not outside the application of these techniques. From the point of view of the modeling of human movement in relation to the learning of complex motor skills, martial arts are of interest because they are articulated around a system of movements that are predefined, or at least, bounded, and governed by the laws of Physics. Their execution must be learned after continuous practice over time. Literature suggests that artificial intelligence algorithms, such as those used for computer vision, can model the movements performed. Thus, they can be compared with a good execution as well as analyze their temporal evolution during learning. We are exploring the application of this approach to model psychomotor performance in Karate combats (called *kumites*), which are characterized by the explosiveness of their movements. In addition, modeling psychomotor performance in a *kumite* requires the modeling of the joint interaction of two participants, while most current research efforts in human movement computing focus on the modeling of movements performed individually. Thus, in this work, we explore how to apply a pose estimation algorithm to extract the features of some predefined movements of *Ippon Kihon kumite* (a one-step conventional assault) and compare classification metrics with four data mining algorithms, obtaining high values with them.

## 1. Introduction

Human activity recognition (HAR) techniques have proliferated and focused on recognizing, identifying and classifying inputs through sensory signals, images or video, and are used to determine the type of activity that the person being analyzed is performing [1]. Following [1], human activities can be classified according to: (i) gestures (primitive movements of the body parts of a person that may correspond to a particular action of this person [2]); (ii) atomic actions (movements of a person describing a certain motion that may be part of more complex activities [3]); (iii) human-to-object or human-to-human interactions (human activities that involve two or more persons or objects [4]); (iv) group actions (activities performed by a group of persons [5]); (v) behaviors (physical actions that are associated with the emotions, personality, and psychological state of the individual [6]); and (vi) events (high-level activities that describe social actions between individuals and indicate the intention or the social role of a person [7].

HAR-type techniques are usually divided into two main groups [5,8]: (i) HAR models based on image or video, and (ii) those that are based on signals collected from accelerometer, gyroscope, or other sensors. Due to legal and other technical issues [5], HAR sensor-based systems are increasing their number of projects. The scope of the sensors at present is large, with cheap and good quality sensors to be able to develop any system in any sector or specific field. In this way, HAR-type systems can gather data from diverse type of sensors such as: accelerometers (e.g., see [9,10]), gyroscopes, which are usually combined with an accelerometer (e.g., see [11,12]), GPS (e.g., see [13,14]), pulse-meters (e.g., see [15,16]), magnetometers (e.g., see [17,18]) and thermometers (e.g., see [19]). As analyzed in [20], the field is still emerging, and inertial-based sensors such as accelerometers and gyroscopes can be used to (i) recognize specific motion learning units and (ii) assess learning performance in a motion unit.

In turn, the models developed based on images and video can provide more meaningful information about the movement, as they can record the movements performed by the skeleton of the body. It is an interdisciplinary technology with a multitude of applications at a commercial, social, educational and industrial level. It is applicable to many aspects of the recognition and modeling of human activity, such as medical, rehabilitation, sports, surveillance cameras, dancing, human–machine interfaces, art and entertainment, and robotics [21,22,23,24,25,26,27,28,29]. In particular, gesture and posture recognition and analysis is essential for various applications such as rehabilitation, sign language, recognition of driving fatigue, device control and others [30].

The world of physical exercise and sport is susceptible to the application of these techniques. What is called the Artificial Intelligence of Things (AIoT) has already emerged, applicable to different sports [31]. In sports and exercise in general, both sensors and video processing are being applied to improve training efficiency [32,33,34] to develop sport and physical exercise systems. Regarding inertial sensor data, [35] provides a systematic review of the field, showing that sensors are being applied in the vast majority of sports, both worn by athletes and in sport tools. Techniques based on computer vision for Physical Activity Recognition (PAR) have mainly used [36]: red-green-blue (RGB) images, optical flow, 2D depth maps, and 3D skeletons. They use diverse algorithms, such as Naive Bayes (NB) [37], Decision Trees (DT) [38], Support Vector Machines (SVM) [39], Nearest Neighbor (NN) [40], Hidden Markov Models (HMM) [41], and Convolutional Neural Networks (CNN) [42]. 

However, currently, hardly any sensor-based or video-based HAR systems address the problem of modeling the movements performed from a psychomotor perspective [43,44], such that they can be compared with the same user along time or with users of different levels of expertise. This would allow to provide personalized guidance to the user to improve the execution of the movements, as proposed in the sensing-modeling-designing-delivering (SMDD) framework described in [45], which can be applied along the lifecycle of technology-based educational systems [46]. In [45], two fundamental challenges are pointed out: (1) modeling psychomotor interaction and (2) providing adequate personalized psychomotor support. One system that follows the SMDD psychomotor framework is KSAS, which uses the inertial sensors from a smartphone to identify wrong movements in a sequence of predefined arm movements in a blocking set of American Kenpo Karate [47]. In turn, and also following the SMDD psychomotor framework, we are developing an intelligent infrastructure called KUMITRON that simultaneously gathers both sensor and video data from two karate practitioners [48,49] to offer expert advice in real time for both practitioners on the Karate combat strategy to follow. KUMITRON can also be used to train motion anticipation performance in Karate practice, based on improving peripheral vision, using computer vision filters [50].

The martial arts domain is useful to contextualize the research on modeling psychomotor activity since martial arts require developing and working on psychomotor skills to progress in the practice. In particular, there are several challenges for the development of intelligent psychomotor systems for martial arts [51]: (1) improve movement modeling and, specifically, movement modeling in combat, (2) improve interaction design to make the virtual learning environment more realistic (where available), (3) design motion modeling algorithms that are sensitive to relevant motion characteristics but insensitive to sensor inaccuracies, especially when using low cost wearables, and (4) create virtual reality environments where realistic force feedback can be provided.

From the point of view of the modeling of human movement in relation to the learning of complex motor skills, martial arts are of interest because they are articulated around a system of movements that are predefined, or at least, bounded, and governed by the laws of Physics [52]. To understand the impact of Karate on the practitioner, there is a fundamental reading by its founder [53]. The physical work and coordination of movements can be practiced individually by performing forms of movements called “*katas*”, which are developed to train from the most basic to the most advanced movements. In turn, the Karate combat (called “*kumite*”) is developed by earning points on attack techniques that are applied to the opponent in combat. The different scores that a karateka (i.e., Karate practitioner) can obtain are: *Ippon* (3 points), *Wazari* (2 points) and *Yuko* (1 point). The criterion for obtaining a certain point in *kumite* is conditioned by several factors, among others, that the stroke is technically well executed [54]. This is why karatekas must polish their fighting techniques to launch clear winning shots that are rewarded with a good score. The training of the combat technique can be performed through the *katas* (individual movements that simulate fighting against an imaginary opponent), or through *Kihon kumite* exercises, where one practitioner has to apply the techniques in front of a partner, with the pressure that this entails.

There are some works that develop and apply movement modeling to the study of the technique performed by karatekas, but only individually [55,56,57]. However, in this work, we are exploring if it is also feasible to develop and apply computer vision techniques to other types of exercises in which the karateka has to apply the techniques with the pressure of an opponent, such as in *Kihon kumite* exercises. Thus, in the current work, we focus on how to model the movements performed in a *kumite* by processing video recordings obtained with KUMITRON when performing the complex and dynamic movements of Karate combats.

In this context, we pose the following research question: “Is it possible to detect and identify the postures within a movement in a Karate *kumite* using computer vision techniques in order to model psychomotor performance?”.

The main objective of this research is to develop an intelligent system capable of identifying the postures performed during a Karate combat so that personalized feedback can be provided, when needed, to improve the performance in the combat, thus, supporting psychomotor learning with technology. The novelty of this research lies in applying computer vision to an explosive activity where an individual interacts with another while performing rapid and strong movements that change quickly in reaction to the opponent’s actions. According to the review of the field in human movement computing that is reported next, computer vision has been applied to identify postures performed individually in sports in general and karate in particular, but we have not found postural identification in the interaction of two or more individuals performing the same activity together. 

Thus, our research seeks to offer a series of advantages in the learning of martial arts, such as studying the movements of practitioners in front of opponents of different heights, in real time and in a fluid way. It is intended to use the advances of this study to improve the technique of Karate practitioners applying explosive movements, which is essential for the assimilation of the technique and hence, achieve improvement in the performance of the movements, according to psychomotor theories [58].

The rest of the paper is structured as follows. Related works are in Section 2. In Section 3, we present the methodology and define the dataset to answer the research question. In Section 4, we explain how the current dataset is obtained. After that, in Section 5 we analyze the dataset and present the results. In Section 6, we discuss the results and suggest some ideas for future work. Finally, conclusions are in Section 7.

## 2. Related Works

The introduction of new technologies and computational approaches is providing more opportunities for human movement recognition techniques that are applied to sports. The extraction of activity data, together with their analysis with data mining algorithms is making the training efficiency higher. Through the acronym HPE (Human Pose Estimation), studies have been found in which different technologies are applied that seek to model the movement of people when performing physical activities.

Human modeling technologies have been applied for the analysis of human movements applied to sport as described in [59]. In particular, different computer vision techniques can be applied to detect athletes, estimate pose, detect movements and recognize actions [60]. In our research, we focus on movement detection. Thus, we have reviewed works that delve into existing methods: skeleton based models, contour-based models, etc. As discussed in [61,62,63], 3D computer vision methodologies can be used for the estimation of the pose of an athlete in 3D, where the coordinates of the three axes (x, y, z) are necessary, and this requires the use of depth-type cameras, capable of estimating the depth of the image and video received.

Martial arts, similar to sport in general, is a discipline where human modeling techniques and HPE are being applied [64,65,66]. Motion capture (mocap) approaches are sometimes used and can add sensors to computer vision for motion modeling, combining the interpretation of the video image with the interpretation of the signals (accelerometer, gyroscope, magnetometer, GPS). In any case, the pose estimation work is usually performed individually [67,68], while the martial artist performs the techniques by themselves.

Virtual reality (VR) techniques have been used to project the avatar or the image of martial artists to a virtual environment programmed in a video game for individual practice. For instance, [69] uses computer vision techniques and transports the individual to the monitor or screen making use of background subtraction techniques [70]. In turn, [71] also creates the avatar of the person to be introduced into the virtual reality and, thus, be able to practice martial arts in front of virtual enemies that are watched through VR glasses, but in this case, body sensors are used to monitor the movements.

The application of computer vision algorithms in the martial arts domain not only focus on the identification of the pose and the movement, but there are advances in the prediction of the next attack. For this, the trainer can be monitored using residual RGB and CNN network frames to which LSTM neural networks are applied that predict the next attack movement in 2D [72]. Moreover, computer vision together with sensor data can be used to record audiovisual teaching material for Physics learning from the interaction of two (human) bodies as in Phy+Aik [73], where Aikido techniques practiced in pairs are monitored and used to show Physics concepts of circular motion when applying a defensive technique to the attack received.

Karate [74] is a popular martial art, invited at the Tokyo Olympics, and thus, there have been efforts in applying new technologies to its modeling from a computing perspective to improve psychomotor performance. In this sense, [75] has reviewed the technologies used in twelve articles to analyze the *“mawasi geri”* (side kick) technique, finding that several kinds of inputs, such as 3D video image, inertial sensors (accelerometers, gyroscopes, magnetometers) and EMG sensors can be used to study the speed, position, movements of body parts, working muscles, etc. There are also studies on the *“mae geri”* movement (forward kick) using the Vicon optical system (with twelve MX-13 cameras) to create pattern plots and perform statistical comparison among the five expert karatekas who performed the technique [76]. Sensors are also used for the analysis of Karate movements (e.g., [77,78]) using Dynamic Time Warping (DTW) and Support Vector Machines (SVM). In this context, it is also relevant to know that datasets of Karate movements (such as [79]) have been created for public use in other investigations as in the above works, but they only record individual movements.

Kinematics has also been used to analyze intra-segment coordination due to the importance of speed and precision of blows in Karate as in [80], where a Vicon camera system consisting of seven cameras (T10 model) is used. These cameras capture the markers that the practitioners wear, and which are divided into sub-elite and elite groups. In this way, a comparison of the technical skill among both groups is made. Using a gesture description language classifier and comparing it with Markov models, different Karate techniques individually performed are analyzed to determine the precision of the model [57].

In addition to modeling the techniques performed individually, some works have focused on the attributes needed to improve the performance in martial arts practice. For instance, VR glasses and video have been used to improve peripheral vision and anticipation [69,81], which has also been explored in KUMITRON [50] with computer vision filters.

The conclusion that we reached after the literature review carried out is that new technologies are being introduced in many sectors, including sports, and hence, martial arts are not an exception. The computation approaches that can be applied vary. In some cases, the signals obtained from sensors are processed, in others computer vision algorithms are used, and sometimes both are combined. In the particular case of Karate, studies are being carried out, although mainly focus on analyzing how practitioners perform the techniques individually. Thus, an opportunity has been identified to study if existing computational approaches can be applied to model the movements during the joint practice of several users, where in addition to the individual physical challenge of making the movement in the correct way, other variables such as orientation, fatigue, adaptation to the anatomy of another person, or the affective state can be of relevance. Moreover, none of the works reviewed uses the modeling of the motion to evaluate the psychomotor performance such that appropriate feedback can be provided when needed to improve the technique performed. Since computer vision seem to produce good results for HPE, in this paper we are going to explore existing computer vision algorithms that can be used to estimate the pose of karatekas in a combat, select one and use it on a dataset created with some *kumite* movements aimed to address the research question raised in the introduction.

## 3. Materials and Methodology

In this section, we present some computer vision algorithms that can be used for pose estimation and present the methodology proposed to answer the research question.

### 3.1. Materials

After the analysis of the state of the art, we have identified several computer vision algorithms that produce good results for HPE, and which are listed in Table 1. According to the literature, the top three algorithms from Table 1 offering best results are WrnchAI, OpenPose and AlphaPose. Several comparisons with varied conclusions have been made among them. According to LearnOpenCV [82], WrnchAI and OpenPose offer similar features, although WrnchAI seems to be faster than OpenPose. In turn, [83] concludes that AlphaPose is above both when used for weight lifting in 2D vision. Other studies such as [84] find OpenPose superior and more robust when applied to real situations outside of the specific datasets such as MPII [85] and COCO [86] datasets. 

From our own review, we conclude that OpenPose is the one that has generated more literature works and seems to have the largest community of developers. OpenPose has been used in multiple areas: sports [87], telerehabilitation [88], HAR [89,90,91], artistic disciplines [92], identification of multi-person groups [93], and VR [94]. Thus, we have selected OpenPose algorithm for this work due to the following reasons: (i) it is open source, (ii) it can be applied in real situations with new video inputs [84], (iii) there is a large number of projects available with code and examples, (iv) it is widely reported in scientific papers, (v) there is a strong developers community, and (vi) the API gives users the flexibility of selecting source images from camera fields, webcams, and others.

### About OpenPose

OpenPose [102] is a computer vision algorithm proposed by the Cognitive Computing Laboratory at Carnegie Mellon University for the real-time estimation of the shape of the bodies, faces and hands of various people. OpenPose provides 2D and 3D multi-person hotspot detection, as well as a calibration toolbox for estimating specific region parameters. OpenPose accepts many types of input, which can be images, videos, webcams, etc. Similarly, its output is also varied, which can be PNG, JPG, AVI or JSON, XML and YML. The input and output parameters can also be adjusted for different needs. OpenPose provides a C ++ API and works on both CPU and GPU (including versions compatible with AMD graphics cards). The main characteristics are summarized in Table 2.

In this way, we can start by exploring the 2D solutions that OpenPose offers so that it can be used with different plain cameras such as the one in a webcam, a mobile phone or even the camera of a drone (which is used in KUMITRON system [48]). Applying 3D will require the use of depth cameras.

As described in [102], OpenPose algorithm works as follows:Deep learning bases the estimation of pose on variations of Convolutional Neural Networks (CNN). These architectures have a strong mathematical basis on which these models are built:
(1)f∗g ≝ ∫−∞∞fτgt−τdτ=∫−∞∞ft−τgτdτApply ReLu (REctified Linear Unit): the rectifier function is applied to increase the non-linearity in the CNN.Group: It is based in spatial invariance, a concept in which the location of an object in an image does not affect the ability of the neural network to detect its specific characteristics. Thus, the clustering allows CNN to detect features in multiple images regardless of the lighting difference in the pictures and the different angles of the images.Flattening: Once the grouped featured map is obtained, the next step is to flatten it. The flattening involves transforming the entire grouped feature map matrix into a single column which is then fed to the neural network for processing.Full connection: After flattening, the flattened feature map is passed through a network neuronal. This step is made up of the input layer, the fully connected layer, and the output layer. The output layer is where the predicted classes are provided. The final values produced by the neural network do not usually add up to one. However, it is important that these values are reduced to numbers between zero and one, which represent the probability of each class. This is the role of the Softmax function.
σ:ℝk→0,1k
(2)σzj=ezj∑1⇐1kezk forj=1,…..,K.All these steps can be represented by the following diagram in Figure 1.

### 3.2. Methodology

The objective of the research introduced in this paper is to determine if computer vision algorithms (in this case, OpenPose) are useful to identify the movements performed in a Karate *kumite* by both practitioners. To answer the question, a dataset will be prepared following predefined *Kihon Kumite* movements both corresponding to attack and defense as explained in Section 4.2. The processing of the dataset consists of two clearly differentiated parts: (i) the training of the classification algorithms of the system using the features extracted with the OpenPose algorithm, and (ii) the application of the system to the input of non-preprocessed movements (raw data). These two stages are broken down into the following sub-stages, as shown in Figure 1:**Acquisition of data input:** Record the movements to create the dataset to be used in the experiment, and later to test it.**Feature extraction**: Applying the OpenPose algorithm to the dataset to group anatomical positions of the body (called keypoints) into triplets to calculate the angle, which allow generating a pre-processed input data file for algorithm training (see Figures 3 and 4). OpenPose allows to have the 2D position of each point (x, y). Thus, by having the coordinates of three consecutive points, the angle formed by those three with respect to the central one is calculated. An example is provided next.**Train a movement classifier:** With the pre-processed data from point 2, train data mining algorithms to classify and identify the movements. Several data mining algorithms can be used for the classification in the current work generating the corresponding learning models (see below). For the evaluation of the classification performance, 10-fold cross validation is proposed, following [105,106,107].**Test the movement classifier:** Apply the trained classifier to the non-preprocessed input (raw data) with the movements performed by the karateka.**Evaluate the performance of the classifiers:** Compare the results obtained by each algorithm in the classification process with usual machine learning classification metrics, currently those offered by Weka: (i) true positive rate, (ii) false positive rate, (iii) precision (number of true positives that are actually positive compared to the total number of predicted positive values), (iv) recall (number of true positives that the model has classified based on the total number of positive values), (v) F-measure (metric that combines precision and recall), (vi) MMC (Mahalanobis Metric for Clustering: minimizes the distances between similarly labeled inputs while maximizing the distances between differently labeled inputs), ROC area (area under the Receiver Operating Curve: used for classification problems and represents the percentage of true positives against the ratio of false positives), and PRC area (Precision Recall Curve: a plot of precision vs. recall for all potential cut-offs for a test).

This process follows the scheme in Figure 2, which shows the steps proposed for Karate movement recognition using OpenPose algorithm to extract the features from the images and data mining algorithms on these features to classify the movements:

OpenPose works with 25 keypoints (numbered from 0 to 24, as shown in Section 4.3 in Figure 9) for the extraction of input data, which are used to generate triples, as it can be seen in Figure 3. OpenPose keypoints are grouped into triplets of three consecutive points, which are the input attributes (angles) for classifying the output classes. To define the keypoints, OpenPose expanded the 18 keypoints of COCO dataset (https://cocodataset.org/#home accessed on 2 December 2021) with the ones for the feet and waist from the Human Foot Keypoint dataset (https://cmu-perceptual-computing-lab.github.io/foot_keypoint_dataset/ accessed on 2 December 2021)). 

The 26 triplets that can be obtained from the 25 keypoints in Figure 3 are grouped and named by their main anatomical part, as shown in Figure 4. 

This allows identifying the numbering of the triples with their body position. As an example we have the keypoints numbered 5, 6 and 7 that correspond to the main part “Shoulders Left Arm” as it can be seen in Figure 4. In Figure 3, the angular creation of the commented triplet can be graphically observed. Figure 4 collects all the triples formed by the keypoints in Figure 3, naming them the main body part they represent. The grouping of the keypoints in triples makes it possible to avoid the variability of the points depending on the height of the practitioners, in order to calculate their angle. In this way, a person of 190 cm tall and another of 170 cm tall, will have similar angles for the same Karate posture, compared to the bigger variation of the keypoints due to the different height of the bodies. This allows the algorithm training to be more efficient and fewer inputs are required to achieve optimal computational performance.

As a result, Figure 5 shows the processing flow proposed to answer the research question regarding the identification of *kumite* postures. On the top part of the figure, the data models are trained to identify all the movements selected for the study. This is done by generating a dataset with the angles of each Karate technique, which is trained according to different data mining algorithms. Subsequently, these trained algorithms are used on real *kumite* movements in real time in order to identify the *kumite* posture performed, as shown on the bottom.

For the selection of the data mining classifiers, a search was made of the types of classifiers applied in general to computer vision. Some works such as [108,109,110,111,112] use machine learning (ML) algorithms, mainly decision trees and bayesian networks. However, deep learning (DL) techniques are more and more used for the identification and qualification of video images as in [113,114,115,116]. In particular, a deep learning classification algorithm that is having very good results according to the studies found is the Weka DeepLearning4j algorithm [117,118,119]. The Weka Deep Learning kit allows to create advanced Neural Network Layers as it is explained in [120].

In addition to a general analysis of ML and DL algorithms, specific application to sports, human modeling and video image processing were also sought. The BayesNet algorithm has already been used to estimate human pose in other similar experiments [121,122,123]. Another algorithm that has been applied to this type of classifications is the J48 decision tree [124,125,126]. These two ML algorithms were selected for the classification of the movements, as well as two neural network algorithms included in the WEKA application: the MultiLayerPerceptron (MLP) algorithm, which has also been used in different works as well as similar algorithms [127,128], and the aforementioned DeepLearning4J. Thus, we have selected two ML and two DL algorithms. Table 3 compares both types of algorithms following [129].

Due to the type of hardware that was used, which consisted of standard cameras (without depth functionality) available in webcams, mobiles and drones, the research was oriented to the application of the video image in 2D format for its correct processing and postural identification.

### 3.3. Computational Cost

The operation of OpenPose for the application programmed consists of a communication system between OpenPose and the functions designed for the video manipulation, done with the four classes in a Java application listed below. The computational cost associated with the processing is determined through the computational analysis of the following main parts of the code that are represented in Figure 6.

The video input is captured by the OpenPose application libraries that identify the keypoints of each individual that appears in the image. As explained in the official OpenPose paper [102], the runtime of OpenPose consists of two major parts: (1) CNN processing time whose complexity is O(1), constant with varying number of people; and (2) multi-person parsing time, whose complexity is O(N^2^), where N represents the number of people.

OpenPose communicates with our application by sending information about the identified keypoints and the number of individuals in the image. The Java application, receives the data from OpenPose, dimensioned in “N” the problem for the complexity calculation. 

There are four main classes:*pxStreamReadCamServer*: receives MxN pixels image compressed in JPEG: (640 * 480) = *O*(*ni*) to decompress*receiveArrayOfDouble*: receives 26 double (angle) = 26 = K2*stampMatOnJLabel*: display image on a Jlabel =~ K1*wrapperClassifier.prediceClase*: deduce position of the angles, 26 * num layers = 26 ∗ 16 = K3

Thus, the cost estimate for these functions is:Cost: *O*(*ni*) + K2 = 26 => 1 (one) + K3 = 26 ∗ 16 => 16 == *O*(*ni*) + *K*;

Therefore, the computational cost of the intelligent movement identification system for an individual and several is: Full cost per frame for one person: 2 ∗ *O*(*ni*) + *K*;Full cost per frame for more than one person: *O*(*N*^2) + 2 ∗ *O*(*ni*) + *K*;

It can be seen how in this case the higher computational cost of the OpenPose algorithm marks the total operational cost of the set of applications, finally reaching a quadratic order.

## 4. Dataset Construction

### 4.1. Defining the Dataset Inputs

To answer the research question, first we have to select the video images to generate the dataset with the interaction of the karatekas. There are different forms of *kumite* in Karate, from multi-step combat (to practice) to free *kumite* [130] (very explosive with very fast movements [131]). Multi-step combat (each step is as an attack) is a simple couple exercise that slowly lead the karateka to an increasingly free action, according to predefined rules. It does not necessarily have to be equated with competition: it is more of a pair exercise in which the participant together with a partner develops a better understanding of the psychomotor technique. Participants do not compete with each other but train with themselves. The different types of *kumite* are listed in Table 4, from those with less freedom of movement to free combat.

In order to follow a progressive approach in our research, we started with the *Ippon Kihon k**umite*, which is a pre-established exercise so that it can facilitate the labeling of movements and their analysis. This is the most basic *kumite* exercise, consisting of the conventional one-step assault. We will use it to compare the results obtained from the application of the data mining algorithms on the feature extracted from the videos recorded and processed using the OpenPose algorithm.

### 4.2. Preparing the Dataset

In this first step of our research, the following attack and defense sequences were defined to create the initial dataset, as shown in Figure 7.

The *Ippon Kihon* k*umite* sequence proposed for this study would actually be (i) *Gedan Barai,* (ii) *Oi Tsuki,* (iii) *Soto Uke, and* (iv) *Gyaku Tsuki*. However, to calibrate the algorithm and enrich the dataset, we also took two postures *Kamae* (which is the starting posture), both for attack and defense, although they are not properly part of the *Ippon Kihon kumite*.

Since monitoring the simultaneous interaction of two karatekas in movement is complex, we have made this first dataset as simple as possible to familiarize ourselves with the algorithm, learn and understand the strengths and weaknesses of OpenPose and explore its potential in the human movement computing scenario addressed in this research (performing martial arts combat techniques between two practitioners). We aim at increasing the difficulty and complexity of the dataset gradually along the research. Thus, for this first dataset, we selected direct upper trunk techniques (not lateral or angular) to facilitate the work of OpenPose classification. In addition, simple techniques with the limbs were selected. Thus, when working in 2D, lateral and circular blows that could acquire angles that could be difficult to calculate in their trajectory and execution, are avoided. The sequence of movements in pairs of attack and defense is shown in Table 5.

### 4.3. Implemented Application to Obtain the Dataset 

Once the dataset was prepared, a Java application was developed in KUMITRON to collect the data from the practitioners. In particular, in this initial and exploratory collection of data, a green belt participant was video recorded performing the proposed movements, both attack and defense. The video was taken statically for one minute in which the karateka had to be in one for the predefined postures of Table 5 and Figure 7, and the camera was moved to capture different possible angles of the shot in 2D. In addition, videos were also taken dynamically in which the karateka went from one posture to another and waited 30 seconds in the last one.

Figure 8 shows the interface of the application while recording the *Oi Tsuki* movement performed by the karateka while testing the application. On the left panel, the postures detected are listed chronologically from bottom to top. On the right of the image, the skeleton identified with the OpenPose algorithm is shown on the real image. 

The application can also identify the skeletons of both karatekas when interacting in an *Ippon Kihon kumite*. Figure 9 shows the left practitioner (male, green belt) launching a *Gedan Barai* attack when the one on the right (women, white belt) is waiting in *Kamae*. 

## 5. Analysis and Results

The attributes selected for the classification with the data mining algorithms, as explained in Section 3.2, were the 26 triples obtained from the OpenPose keypoints. In this way, it is possible to calculate the angles that are formed by the different joints of the body of each karateka. Figure 10 shows the 26 different attributes considered corresponding to the 26 triplets introduced in Figure 4. They are shown in order as they are sent by the application. The last window shows the classes (postures) that are being identified. Colors are assigned as indicated in Table 6.

For each of the postures (classes) several videos were recorded with KUMITRON to create the initial dataset (as described in Section 4.3) and obtain the video inputs of the postures for the OpenPose algorithm indicated in Table 6. The postures were recorded in different ways. On the one hand, one minute long videos were made where the camera moved from one side to the other while the karateka was completely still maintaining the posture. To generate postural variability, other videos were created where the karateka performed the selected postures for the *Ippon Kihon kumite*, stopping at the one that was to be recorded. In this way, it is assumed that the recorded videos can offer more difference between keypoints than just making static recordings. From the recorded videos, the inputs generated come out at a rate of 25 frames per second. Thus, from one second of video, 25 inputs are generated for every second. A feature in OpenPose to clean the inputs with values very different from the average ones was used to deal with the members that remain in the back of the camera.

In Figure 10, it can be observed in a range of 180 degrees (range in which the 26 angles move) the values that each class takes according to the part of the body. This can be interesting in future research to investigate, for example, whether a posture is overloaded and is likely to cause some type of injury because it is generating an angle that what it is convenient for that part of the body.

The statistical results obtained by applying the four algorithms (BayesNet, J48, MLP and DeepLearning4J) using Weka suite to the identification of movements of the *Ippon Kihon kumite* dataset built in this first step of our research are presented in Table 7. The dataset obtained for the expression is publicly accessible (https://github.com/Kumitron accessed on 2 December 2021). 

Table 7 shows that the classifying algorithms have obtained a good classification performance without significant differences among them. Nonetheless, the performance of the algorithms will need to be revaluated when more users and more movements are included in the dataset and it becomes more complex. The detail of the evaluation metrics results by type of algorithm is shown in Table 8. 

As expected from Table 7, evaluation metrics are good in all the algorithms. DeepLearning4J seems to perform slightly better. However, another important variable to consider in the performance analysis between algorithms is the time it takes to build the models. Figure 11 shows that, from the comparison of the four algorithms, the differences in processing time between the ML and DL algorithms are high. As expected, neural networks take much longer, and since performance results are similar, they do not seem necessary for the current analysis. However, we will analyze how both the processing time evolves in these algorithms, with a greater number of inputs and varying conditions of speed of movements in future versions of the dataset.

The detailed performance results by type of movement (class) for each algorithm is shown in Table 9, Table 10, Table 11 and Table 12 reporting the values of the metrics computed by Weka. These results were already good in the overall analysis, and no significant difference was found between types of movements in any of the algorithms. 

The metrics computed for the BayesNet algorithm are reported in Table 9.

The metrics computed for the J48 algorithm are reported in Table 10.

The metrics computed for the MLP algorithm are reported in Table 11.

The metrics computed for the BayesNet algorithm are reported in Table 12.

### Network Hyperparameters

For the application of the classification models, possible methods of optimization of the hyperparameters were investigated [132]. Since the results obtained with the current dataset and default parameters in Weka were good, no optimization was performed, and default values were left. Table 13 shows the scripts used to call the algorithms in Weka. 

## 6. Discussion

Recalling the classification of human activities introduced in Section 1 [1], this reesearch focuses on type iv, that is, group activities, which focus on the interaction of actions between two people. That is the line of research that makes this work different from those found in the state of the art. The application of OpenPose algorithm to *kumite* recorded videos aims to facilitate improving the practitioners’ technique against different opponents when applying the movements freely and in real time. This is important for the assimilation of the techniques in all kinds of scenarios, including sport practice and personal defense situations. The modeling of movements for psychomotor performance can provide a useful tool to learn Karate that opens up new lines of research.

To start with, it can allow studying and improving the combat strategy, having exact measurements of the distance necessary to know when a fighter can be hit by the opponent’s blows, as well as the distance necessary to reach the opponent with a blow. This allows training in the gym in a scientific and precise way for combat preparation and distance taking on the mat. In addition, studies can be carried out on how some factors such as fatigue can create variability in the movements developed within a combat in each one of its phases. This is important because this allows for deciding on the choice of certain fighting techniques at the beginning or end of the *kumite* to win. Modeling and capturing tagged movements can also be used to produce datasets and to apply them in other areas such as cinema or video games, which are economically attractive sectors.

The current approach to progress in our research is to label not only single postures (techniques) within the *Ippon Kihon kumite* (as done here) but the complete sequences of the movements. There is some work that can guide the technical implementation of this approahc, where classes are first identified and then become subclasses of a superclass [133]. In the work reported in this paper, classes have been defined as the specific techniques to be identified. In the next step, the idea is that these techniques are considered as subclasses, being part of a superior class or superclass. In this way, when the system identifies a concatenation of specific movements, it should be able to classify the attack/defense sequence (default *Ippon Kihon kumite*) that is being carried out as shown in Figure 12.

This can be useful in order to add personalized support to the *kumite* training based on the analysis of the psychomotor performance both comparing the current movements with a good execution and analyzing the temporal evolution of the execution of the movements. The system will be able to recognize if the karateka is developing the movements of a certain section correctly, not only one technique, but the entire series. Thus, training is expected to help practitioners assimilate the concatenations of movements in attack and defense. Furthermore, having super classes identified by the system makes it possible to know in advance which next subclass will be carried out, in such a way that the following movements that will be carried out by a karateka can be monitored through the system, provided that they follow the pattern of some predefined concatenated movements. This is of special interest to be able to work the anticipation training for the defender.

For this, a methodology has to be followed for creating super classes that bring together the different concatenated movements. In principle, it seems to be technically possible, as it has been introduced above, but more work is necessary to be able to answer the question with absolute security, since in order to recognize a superclass, the application must recognize all subclasses without any errors or identification of any wrong postures in the sequence. This requires high identification and classification precision. Moreover, the system needs to have in memory the different movements of each defined superclass, which means that it knows which movement should be the next to perform. In this way, training can be enriched with attack anticipation work in a dynamic and natural way.

For future work, and in addition to exploring other classification algorithms, such as the LSTM algorithm that have reported good results in other works [47,134], it would be interesting to add more combinations of movements to create a wider base of movements and to compare the results between different *Ippon Kihon kumite* movements. Thus, the importance of the different keypoints from OpenPose will be studied (including the analysis of the movements that are more difficult to identify) as well as if it is possible to eliminate some to make the application lighter. For the *Ippon Kihon kumite* sequence of postures that has been used in the current work, the keypoints of the legs does not seem to be especially important since the labeled movements are few, and the monitored postures are well identified only with the upper part of the body.

Next, the idea is to extend the current research to faster movements that are performed in other types of *kumite*, following the order of difficulty of these, from *Kihon Sambon kumite* to competition *kumite*, or *kumite* with free movements. This will allow the calibration of the technical characteristics of the algorithm used (currently OpenPose) to check what type of image speed is capable of working with while still obtaining satisfactory results. This would also be an important step forward in adding elements to anticipation and peripheral vision training. Such attributes have already been started to be explored in our research using OpenCV algorithms [50].

The progress in the classification of the movements during a *kumite* will be integrated into KUMITRON intelligent psychomotor system [48]. KUMITRON collects both video and sensor data from karatekas’ practice in a combat, models the movement information, and after designing the different types of feedback that can be required (e.g., with TORMES methodology [135]), delivers the appropriate one for each karateka in each situation (e.g., taking into account the karatekas’ affective state during the practice, which can be obtained with the physiological sensors available in KUMITRON following a similar approach as in [136]) through the appropriate sensorial channel (vibrotactile feedback should be explored due to the potential discussed in [137]). In this way, it is expected that computer vision support in KUMITRON can help karatekas learn how to perform the techniques in a *kumite* with the explosiveness required to win the point, making rapid and strong movements that quickly react in real time and in a fluid way to the opponent’s technique, adapting also the movement to the opponent’s anatomy. The psychomotor performance of the karatekas is to be evaluated both comparing the current movements with a good execution and analyzing the temporal evolution of the execution of the movements.

## 7. Conclusions

The main objective of this work was to carry out a first step in our research to assess if computer vision algorithms allow identifying the postures performed by karatekas within the explosive movements that are developed during a *kumite*. The selection of *kumite* was not accidental. It was chosen because it challenges image processing in human movement computing, and there is little scientific literature on modeling the psychomotor performance in activities that involve the joint participation of several individuals, as in a karate combat. In addition, the movements performed by a karateka in front of an opponent may vary with respect to performing them alone through *katas* due to factors such as fear, concentration or adaptation to the opponent’s physique (e.g., height).

The results obtained from the training of the classification algorithms with the features extracted from the recorded videos of different *Kihon kumite* postures and their application to non labelled images have been satisfactory. It has been observed that the four algorithms used to classify the features extracted with OpenPose algorithm for the detection of movements (i.e., BayesNet, J48, MLP and DeepLearning4j in Weka) have a precision of above 99% for the current (and limited) dataset. With this percentage of success, it is expected that progress can be made with the inclusion of more *Kihon kumite* sequences in a new version of the dataset to analyze whether they can be identified within a *Kihon kumite* fighting exercise. In this way, the research question would be satisfied for the training level of *Kihon kumite*. 

## Figures and Tables

**Figure 1 sensors-21-08378-f001:**
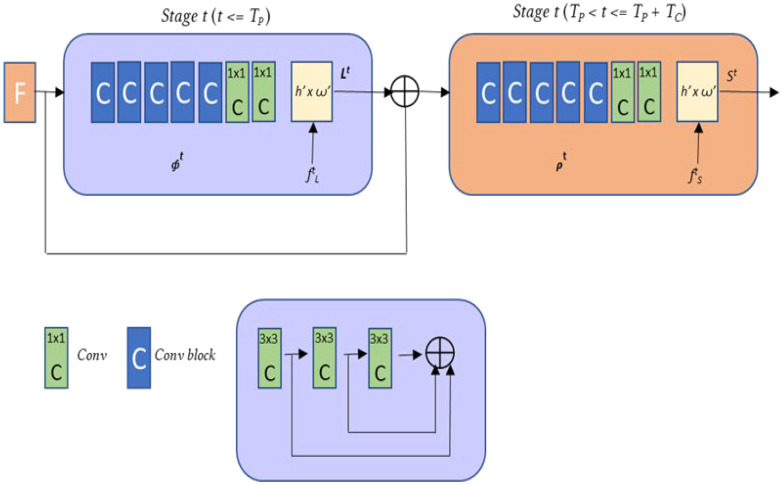
OpenPose’s network structure schematic diagram (obtained from [104]).

**Figure 2 sensors-21-08378-f002:**
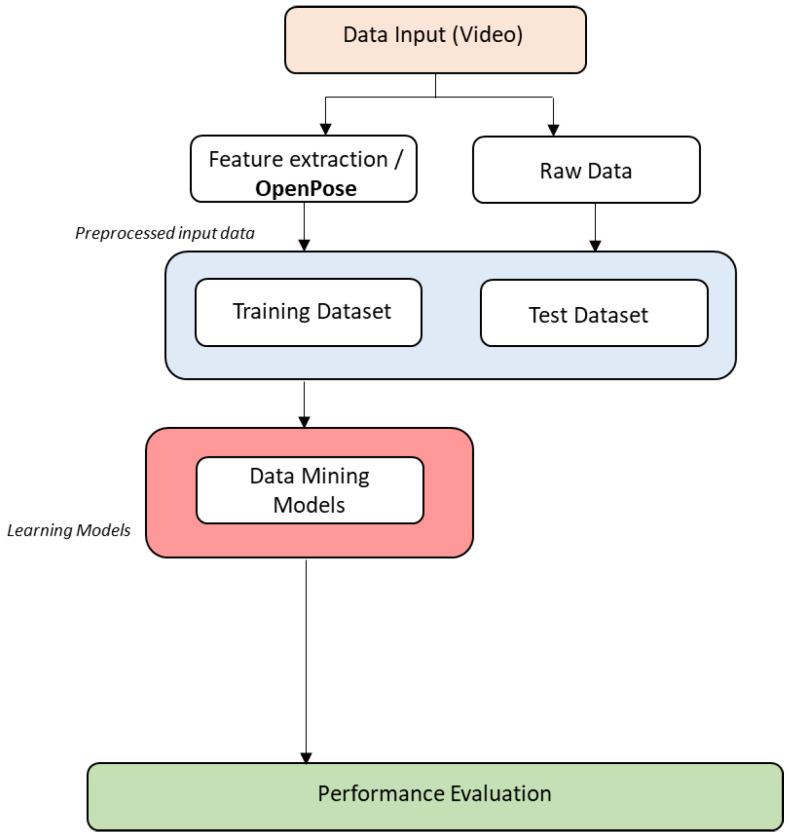
Steps for Karate movement recognition using OpenPose.

**Figure 3 sensors-21-08378-f003:**
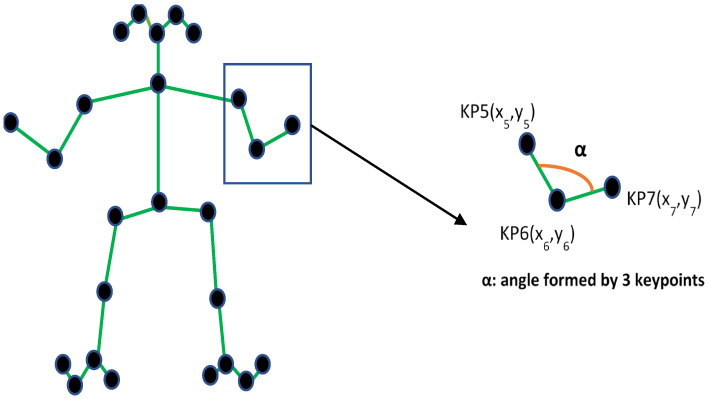
Example of angle feature extraction from the keypoints obtained by combining COCO and Human Foot Keypoint datasets.

**Figure 4 sensors-21-08378-f004:**
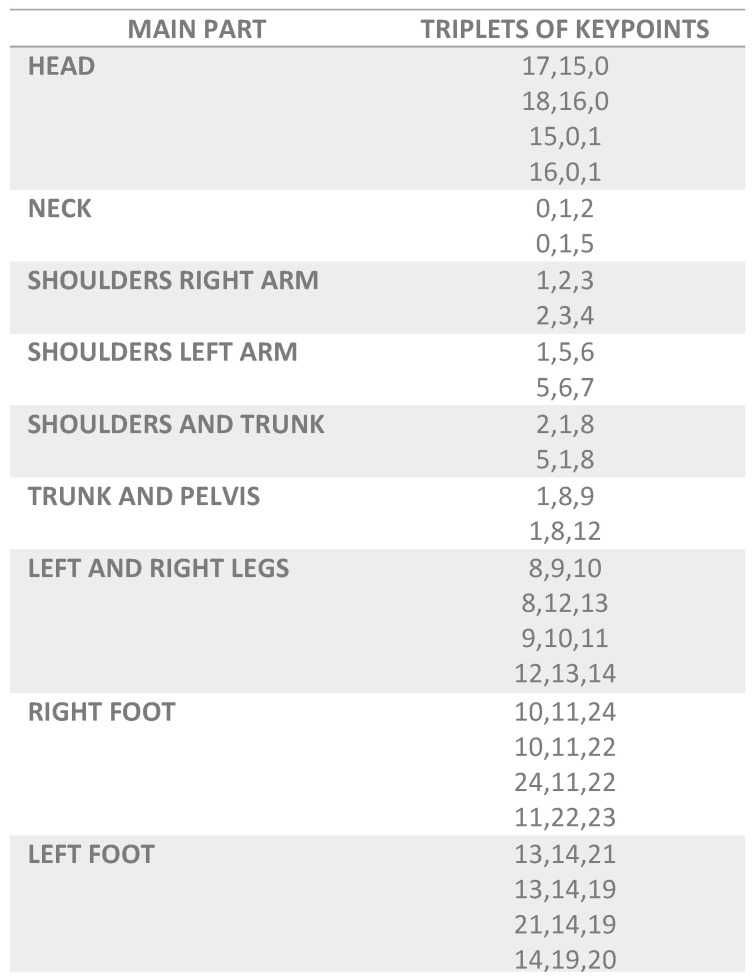
Triplets of keypoints for calculating the angle with OpenPose.

**Figure 5 sensors-21-08378-f005:**
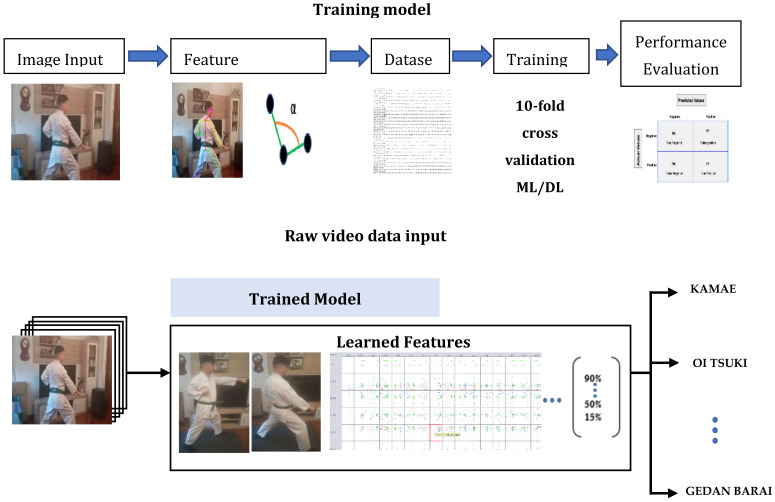
Training model and the raw data input process flow proposed in this research.

**Figure 6 sensors-21-08378-f006:**
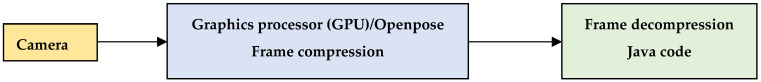
Process to calculate the computational cost.

**Figure 7 sensors-21-08378-f007:**
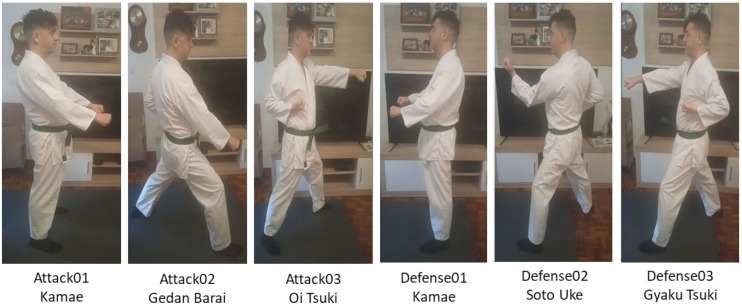
Pictures of the postures in an *Ippon Kihon kumite* selected for the initial dataset.

**Figure 8 sensors-21-08378-f008:**
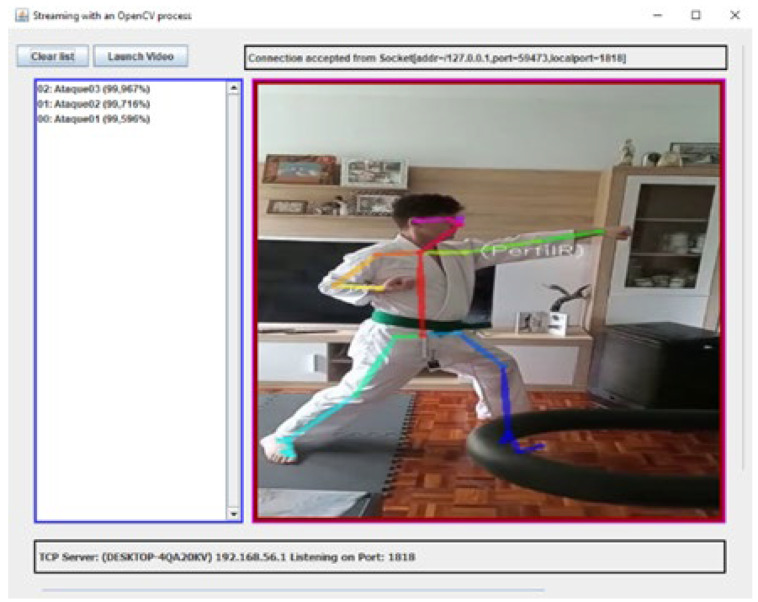
Performing individually an *Ippon Kihon kumite* movement (*Oi Tsuki* attack) for the recognition tests.

**Figure 9 sensors-21-08378-f009:**
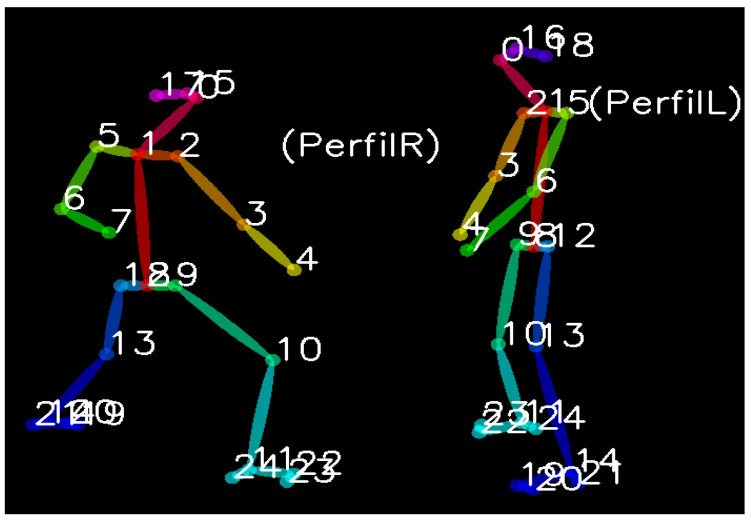
Applying OpenPose to *Ippon Kihon kumite* training of two participants, where the body keypoints are represented (image obtained from KUMITRON).

**Figure 10 sensors-21-08378-f010:**
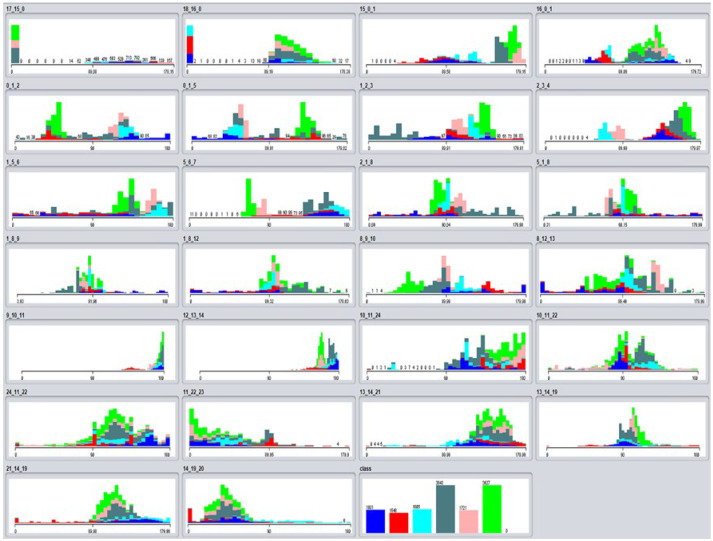
Display of defined attributes selected for identification in defined classes (obtained from Weka).

**Figure 11 sensors-21-08378-f011:**
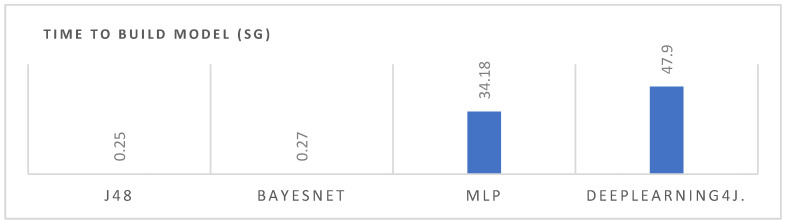
Comparative times between the applied algorithms to build the models.

**Figure 12 sensors-21-08378-f012:**
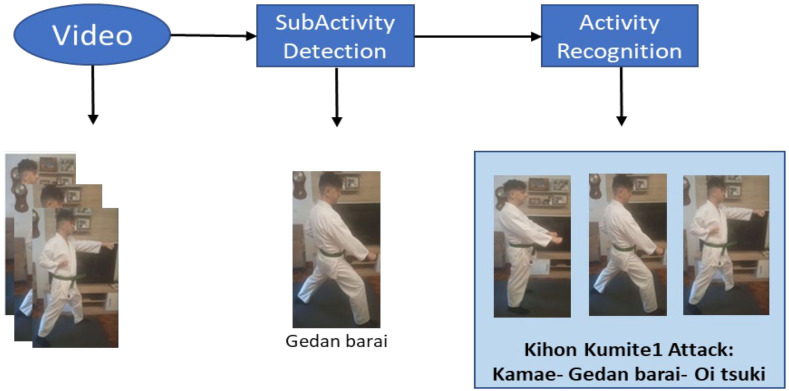
Subactivity recognition in a sequence of *Ippon Kihon kumite* techniques.

**Table 1 sensors-21-08378-t001:** Description of existing computer vision algorithms for pose estimation.

Pose Estimation Algorithms	Description
**AlphaPose** (https://github.com/MVIG-SJTU/AlphaPose, accessed on 28 November 2021)	Presented in 2016 [95], it is an algorithm that allows estimating the pose of one or more individuals. It is the first open source system that has reached the following records: 80+ mAP (82.1 mAP) on MPII dataset and 70+ mAP (72.3 mAP) on COCO dataset. This means that the algorithm is more precise in detecting keypoints in comparison with others. AlphaPose is free to use and distribute as long as it is not used for commercial purposes.
**DeepCut** (https://github.com/eldar/deepcut, accessed on 28 November 2021)	System developed in 2016 [96] presented as a multi-person computer vision system, with deeper, stronger and faster features compared to the state of the art at that time. It works bottom-up for image treatment. The way of working is to detect the people who are in an image to later predict the joint locations. It can be applied to both images and video of sports such as baseball, athletics or soccer.
**Deep Pose** (https://github.com/mitmul/deeppose, accessed on 28 November 2021)	An algorithm presented in 2014 [97] that estimates the human pose using Deep Neural Networks (DNN). To do this, a regression based on DNN is performed to estimate the joints. In challenges of precision in the classification of images [98], DeepPose obtained better results than the rest of the works, becoming a benchmark of that moment.
**DensePose** (https://github.com/facebookresearch/DensePose, accessed on 28 November 2021)	It is an algorithm developed in 2018 by members of Facebook [99] that maps the pixels of the human body in 2D to turn it into a 3D surface that covers the human body. It serves one or more individuals. It is being used to determine the surface of the human body for different purposes such as trying on virtually an article of clothing on the avatar created for oneself.
**High Resolution Net (HRNet)** (https://github.com/HRNet/HigherHRNet-Human-Pose-Estimation, accessed on 28 November 2021)	Neural network architecture for the estimation of human pose developed in 2019 by Microsoft [100]. It is also used for semantic segmentation and object detection. Despite being a relatively new model, it is becoming a benchmark in the field of computer vision algorithms. It has been the winner in several computer vision tournaments, for example in ICCV2019 [101]. It is a useful architecture to implement in the postural analysis of televised events since it makes high-resolution estimates of postures.
**OpenPose** (https://github.com/CMU-Perceptual-Computing-Lab/openpose, accessed on 28 November 2021)	Computer vision algorithm for the estimation of pose in real time of several people in 2D developed in 2017 [102]. It has undergone functionalities extensions, and currently allows to be used in 3D, hand point detection, face detection, and work with Unity. The OpenPose API allows obtaining the image from various devices: recorded video, streaming video, webcam, etc. Other hardware is also supported, such as CUDA GPUs, OpenCL GPUs, and CPU-only devices.
**PoseNet** (https://github.com/tensorflow/tfjs-models/tree/master/posenet, accessed on 28 November 2021)	It is a pose estimator for a single person or several people, offering 17 keypoints with which to model the human body. It was developed in 2015 [103]. At first, it was aimed at lightweight devices such as mobile phones or browsers, although today it has advanced and improved performance.
**WrnchAI** (https://go.hingehealth.com/wrnch, accessed on 28 November 2021)	WrnchAI is a human deposit estimation algorithm developed by a company based in Canada in 2014 and released only under license. It can be used for one or several individuals making use of the low latency engine, being a system compatible with all types of video. Due to its commercial use, we could not find any scientific paper describing it.

**Table 2 sensors-21-08378-t002:** Technical characteristics of the OpenPose algorithm (obtained from https://github.com/CMU-Perceptual-Computing-Lab/openpose, accessed on 28 November 2021).

	OpenPose Features
Main functionality (with a plain camera)	Detection of keypoints of several people in real time 2D. Body/foot keypoint estimate of 15, 18 or 25 keypoints, including 6 foot keypoints. Execution time invariable with respect to the number of people detected. Handheld keypoint estimate of 2 × 21 keypoints. The execution time depends on the number of people detected. Estimation of keypoints of faces of 70 keypoints. The execution time depends on the number of people detected.
Real-time single-person 3D keypoint detection	3D triangulation of multiple unique views. Synchronization of Flir cameras managed. Compatible with Flir/Point Gray cameras.
Calibration Toolbox	Estimation of the distortion, intrinsic and extrinsic parameters of the camera.
Input	Image, Video, Webcam, Flir/Point Gray, IP Camera, and support for adding your own custom input source (e.g., depth camera).
Output	Basic image + keypoint display/save (PNG, JPG, AVI...), keypoint save (JSON, XML, YML...), keypoints as array class and support to add your own code custom output (e.g., some fancy user interface).

**Table 3 sensors-21-08378-t003:** Machine Learning vs. Deep Learning (obtained from [129]).

Characteristics	Machine Learning	Deep Learning
**Data Requirement**	Small/Medium	Large
**Accuracy**	High accuracy	Medium accuracy
**Preprocessing phase**	Needed	Not needed
**Training time**	Short time	Takes longer time
**Interpretability**	From easy (tree, logistic) to difficult (SVM)	From difficult to impossible
**Hardware requirement**	Trains on CPU	Requires GPU

**Table 4 sensors-21-08378-t004:** Types of *kumite*.

*KIHON*-*KUMITE* (multi-step combat)	*IPPON KIHON KUMITE*: One-step conventional assault.
	*SAMBON KIHON KUMITE*: Three-step conventional assault.
	*GOHON KIHON KUMITE*: Five-step conventional assault.
*KUMITE*	*JYU IPPON KUMITE*: Free and flexible assault one step away. It can have different work types: (i) announcing height and type of attack, (ii) announcing height, (iii) announcing type of attack, and (iv) unannounced.
	*URA IPPON KUMITE* (*Kaisho Ippon Kumite*): Unconventional one-step assault. In this type of work one of the karatekas (acting as *uke*) performs the attack and the other (as *tori*) defends it and counterattacks the *uke* who defends the counterattack by the *tori* and ends up counterattacking. There are three working types: (i) announcing the attack and with the pre-established counterattack, (ii) announcing the attack and with the free counterattack, and (iii) unannounced.
	*JIYU KUMITE*: Free and flexible combat.
	*SHIAI KUMITE*: Regulated combat for competition.

**Table 5 sensors-21-08378-t005:** Postures to be detected in the dataset for the *Ippon Kihon kumite*.

	Attack		Defense
1	Kamae	1	Kamae
1	Gedan Barai	2	Soto Uke
3	Oi Tsuki	3	Gyaku Tsuki

**Table 6 sensors-21-08378-t006:** Defined Karate postures with the colors used in Figure 10 and the number of dataset inputs considered.

Class	Color	Number of Dataset Inputs
**Attack01: “Kamae”**	Blue	1801
**Attack02: “Gedan Barai”**	Red	1548
**Attack03: “Oi Tsuki”**	Cyan	1805
**Defense01: “Kamae”**	Dark green	3643
**Defense02: “Soto Uke”**	Pink	1721
**Defense03: “Gyaku Tsuki”**	Light green	3627

**Table 7 sensors-21-08378-t007:** Summary results of the four data mining algorithms using 10-fold cross validation.

	J48	BayesNet	MLP	DeepLearning4J
**Time to build model (seg)**	0.5	0.27	34.18	47.9
**Correctly Classified Instances**	14139	14137	14138	14142
**% Correct**	99.9576	99.9434	99.9505	99.9788
**Incorrectly Classified Instances**	6	8	7	3
**% Incorrect**	0.0424	0.0566	0.0495	0.0212
**Kappa statistic**	0.9995	0.9993	0.9994	0.9997
**Mean absolute error**	0.0001	0.0002	0.0006	0.0001
**Root mean squared error**	0.0119	0.0137	0.0118	0.0074
**Relative absolute error**	0.0539	0.0702	0.2148	0.0521
**Root relative squared error%**	3.2416	3.7228	3.2134	2.0249
**Total number of Instances**	14145	14145	14145	14145

**Table 8 sensors-21-08378-t008:** Evaluation metrics obtained for the four algorithms.

Algorithm	TP Rate	FP Rate	Precision	Recall	F-Measure	MCC	ROC Area	PRC Area
**BayesNet**	1.000	0.000	0.999	0.999	0.999	0.999	1.000	1.000
**J48**	1.000	0.000	1.000	1.000	0.999	0.999	1.000	0.999
**MLP**	1.000	0.000	1.000	1.000	1.000	0.999	1.000	1.000
**DeepLearning4J**	1.000	0.000	1.000	1.000	1.000	1.000	1.000	1.000

**Table 9 sensors-21-08378-t009:** BayesNet algorithm—metrics computed by movement (class).

	TP Rate	FP Rate	Precision	Recall	F-Measure	MCC	ROC Area	PRC Area	Class
	1.000	0.000	0.999	1.000	0.999	0.999	1.000	1.000	Attack01
	1.000	0.000	0.996	1.000	0.998	0.998	1.000	0.999	Attack02
	0.996	0.000	1.000	0.996	0.998	0.997	1.000	1.000	Attack03
	1.000	0.000	1.000	1.000	1.000	1.000	1.000	1.000	Defense01
	1.000	0.000	1.000	1.000	1.000	1.000	1.000	1.000	Defense02
	1.000	0.000	1.000	1.000	1.000	1.000	1.000	1.000	Defense03
**Avg.**	1.000	0.000	0.999	0.999	0.999	0.999	1.000	1.000	

**Table 10 sensors-21-08378-t010:** J48 algorithm—metrics computed by movement (class).

	TP Rate	FP Rate	Precision	Recall	F-Measure	MCC	ROC Area	PRC Area	Class
	0.999	0.000	0.998	0.999	0.999	0.998	0.999	0.997	Attack01
	0.998	0.000	0.999	0.998	0.998	0.998	0.999	0.997	Attack02
	0.999	0.000	1.000	0.999	1.000	1.000	1.000	1.000	Attack03
	1.000	0.000	1.000	1.000	1.000	1.000	1.000	1.000	Defense01
	1.000	0.000	0.999	1.000	1.000	1.000	1.000	0.999	Defense02
	1.000	0.000	1.000	1.000	1.000	1.000	1.000	1.000	Defense03
**Avg.**	1.000	0.000	1.000	1.000	0.999	0.999	1.000	0.999	

**Table 11 sensors-21-08378-t011:** MLP algorithm—metrics computed by movement (class).

	TP Rate	FP Rate	Precision	Recall	F-Measure	MCC	ROC Area	PRC Area	Class
	0.999	0.000	0.999	0.999	0.999	0.999	0.999	1.000	Attack01
	0.997	0.000	0.998	0.997	0.998	0.997	1.000	1.000	Attack02
	0.999	0.000	0.999	0.999	0.999	0.999	1.000	1.000	Attack03
	1.000	0.000	1.000	1.000	1.000	1.000	1.000	1.000	Defense01
	1.000	0.000	0.999	1.000	1.000	1.000	1.000	0.999	Defense02
	1.000	0.000	1.000	1.000	1.000	1.000	1.000	1.000	Defense03
**Avg.**	1.000	0.000	1.000	1.000	1.000	0.999	1.000	1.000	

**Table 12 sensors-21-08378-t012:** Deep Learning4J—metrics computed by movement (class).

	TP Rate	FP Rate	Precision	Recall	F-Measure	MCC	ROC Area	PRC Area	Class
	0.999	0.000	0.999	0.999	0.999	0.999	0.999	1.000	Attack01
	0.999	0.000	1.000	0.999	0.999	0.999	1.000	1.000	Attack02
	1.000	0.000	0.999	1.000	0.999	0.999	1.000	1.000	Attack03
	1.000	0.000	1.000	1.000	1.000	1.000	1.000	1.000	Defense01
	1.000	0.000	1.000	1.000	1.000	1.000	1.000	0.999	Defense02
	1.000	0.000	1.000	1.000	1.000	1.000	1.000	1.000	Defense03
**Avg.**	1.000	0.000	1.000	1.000	1.000	1.000	1.000	1.000	

**Table 13 sensors-21-08378-t013:** Network hyperparameters using Weka.

Network Hyperparameters	
**BayesNet**	weka.classifiers.bayes.BayesNet -D -Q weka.classifiers.bayes.net.search.local.K2 -- -P 1 -S BAYES -E Weka.classifiers.bayes.net.estimate.SimpleEstimator -- -A 0.5
**J48**	weka.classifiers.trees.J48 -C 0.25 -M 2
**MLP**	weka.classifiers.functions.MultilayerPerceptron -L 0.3 -M 0.2 -N 500 -V 0 -S 0 -E 20 -H a
**DeepLearning4j**	weka.dl4j.inference.Dl4jCNNExplorer -custom-model “weka.dl4j.inference.CustomModelSetup -channels 3 -height 224 -width 224 -model-file C:\\Users\\Johni\\wekafiles\\packages” -decoder “weka.dl4j.inference.ModelOutputDecoder -builtIn IMAGENET -classMapFile C:\\Users\\Johni\\wekafiles\\packages” -saliency-map “weka.dl4j.interpretability.WekaScoreCAM -bs 1 -normalize -output C:\\Users\\Johni\\wekafiles\\packages -target-classes -1” -zooModel “weka.dl4j.zoo.Dl4jResNet50 -channelsLast false -pretrained IMAGENET”

## Data Availability

Not applicable.

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
