# Peer review of "Toward Modeling Psychomotor Performance in Karate Combats Using Computer Vision Pose Estimation"

_sensors, 2021, doi:10.3390/s21248378_

Round 1

Reviewer 1 Report

The problem of pose identification is a very important one.  The paper needs some improvements in my opinion.

The scientific contribution should be very clearly specified. At this stage the paper does a very good literature review as well as the description of the  existing algorithms,  but it does not contribute any novel method. What is probably new is the domain of application i.e.  karate but it is not sufficient for a research paper.  How can you use the results? Have you thought about converting the results to be useful for authomatic generation of poses in games?

Could you also provide some data about computational cost?

Detailed comments

  1. Do not use white text on light blue background in tables, it is very difficult to read online, and i possible to read when printed in black and white. The same about the use of colors, add some other differentiation, for example instead of red line and blue line you can  have red dashed line and blue dotted line.
  2.  Please proofread the paper. Some sentences are hard to understand for example, what do you mean by "document has had the objective of finishing if with" in line 584.

Reviewer 2 Report

The authors propose a modest objective of applying computer vision to analyze karate combat. The paper lacks scientific novelty and does not surprise anyone; computer vision should work on the objective being analyzed in this paper. 

 There are a few minor issues that I want the authors to address. 

  1. Please include the details of the networks (parameter/ hyperparameters) and the dataset used so that the results can be reproduced independently. If possible please share the implementation publicly.
  2. Add some discussion regarding how the current finding can be used to improve real-world human karate performance.

As the paper is well written and looks to be scientifically sound. I propose to accept it with minor modifications. 

Author Response

Thanks for your interesting comments. Please see the attachment.
